

# An efficient propagation system through stem cuttings of a multipurpose plant—*Ficus tikoua* Bur

Tinghong Tan[1,2], Yu Peng[1], Biling An[1], Fan Gao[1], Yanni Sun[3], Chuandong Yang[1,2], Hong Yang[1,2] and Zhihong Lu[4]

[1] School of Agriculture and Forestry Engineering and Planning, Tongren University, Tongren, China
[2] Guizhou Key Laboratory of Biodiversity Conservation and Utilization in the Fanjing Mountain Region, Tongren University, Tongren, China
[3] Tongren No. 1 Middle School of Guizhou, Tongren, China
[4] Guangxi Vocational University of Agriculture, Nanning, China

Corresponding authors
Tinghong Tan, tthsqzx@126.com
Zhihong Lu, luzhihong305@163.com

## ABSTRACT

*Ficus tikoua* Bur., a versatile plant with medicinal, edible, landscaping, and ecological applications, holds significant economic value and boasts a long-standing history of utilization in China. Despite its robust adaptability, rapid growth, and extensive distribution, the current research gap concerning the physiological mechanisms underlying stem cutting propagation hampers the development of efficient strategies for commercial-scale propagation of *F. tikoua*, particularly for large-scale cultivation. To address this, we investigated the effects of habitat heterogeneity, physiological indicators, and environmental factors on the cutting propagation of *F. tikoua*. Stem segments were collected from grassland, sandy, rocky and understory habitats in the field and subjected to comprehensive analyses using a two-factor mixed experimental design and progressive group experiments. Our findings indicate that stem segments of *F. tikoua* with a length of 10 cm and a diameter of 0.5∼0.7 cm exhibited the highest shoot induction rate and total multiplication coefficient. Optimal results were achieved with a vertical burial depth of 5∼7 cm. Stem segments aged 2∼3 years produced the highest number of seedlings, and the most suitable propagation period for cuttings was from March to April. The best substrate-habitat combinations for overall seeding yield were grassland-T4 (loam: light substrate: humus = 2:1:1), sandy-T6 (loam: light substrate: humus = 2:3:1), rocky-T5 (loam: light substrate: humus = 2:2:1), understory-T3 (loam: light substrate: humus = 1:1:1), respectively. Mantel test analyses revealed that the ability of *F. tikoua* cuttings from different habitats to form adventitious roots (ARs) largely determined the functional traits associated with cutting propagation. Overall, our results suggest that stem segment from grassland habitat are the most suitable for *F. tikoua* cutting propagation, with a loam: light substrate: humus ratio of 2:1:1 being the most favorable substrate. In contrast, *F. tikoua* from rocky habitat is not suitable for cutting propagation, particularly for large-scale seedling production.

## INTRODUCTION

*Ficus tikoua* Bur., an evergreen woody creeping vine belonging to the Moraceae family, is commonly found in various habitats such as roadside areas, grassy slopes, sandy lands, river beaches, stone walls, and rocky crevices (*Wang et al., 2016*; *Wang et al., 2020*). Characterized by its strong adaptability and rapid growth, *F. tikoua* is widely distributed across southern China, including provinces like Hunan, Guizhou, Guangxi, Yunnan, and Sichuan, as well as in countries such as India, Vietnam, Thailand, and Laos (*Fu et al., 2018*). This plant has been utilized in traditional folk medicine for centuries to treat ailments such as diarrhea, jaundice, spermatorrhea, cough, edema, and amenorrhea (*Wei, Wu & Ji, 2012*; *Bervinova et al., 2022*). After ripening, its fruits are not only delicious and nutritious but also suitable for all ages, making them a valuable wild fruit.

The vine of *F. tikoua* exhibits strong flexibility, allowing it to grow by creeping, climbing, and hanging. Its buds possess robust sprouting abilities, resulting in abundant shoots, while its leaves remain green all year round, making it an excellent choice for landscaping and ornamental purposes. Additionally, due to its strong rooting ability and tendency to grow in a criss-cross pattern, *F. tikoua* plays a crucial role in preventing soil desertification and soil consolidation, serving as an ideal material for vegetation recovery on highway slopes and park green spaces (*Wang et al., 2019*; *Zhou et al., 2022*). Thus, *F. tikoua* is a multi-functional plant that integrates medicinal, edible, landscaping, and ecological benefits, with significant economic value and broad application prospects (*Wei et al., 2011*; *Shi et al., 2012*; *Zhou et al., 2018*).

In recent years, scholars have conducted extensive research on the chemical composition and stress resistance of *F. tikoua* (*Wu et al., 2015*; *Chai et al., 2017*; *Zhou et al., 2023*). Studies have also explored its landscaping potential, photosynthetic characteristics, and fruit nutrition (*Wei et al., 2012*; *Shi et al., 2012*; *Yuan et al., 2021*; *Chen et al., 2024*). Notably, the chloroplast genome of *F. tikoua* has been sequenced, and transcriptomic data have been analyzed to identify differentially expressed genes in stems and leaves (*Wang et al., 2020*). Furthermore, research has focused on its ecological adaptability and superficial exploitation and utilization technologies (*Liu et al., 2011*; *Ghestem et al., 2014*; *Li et al., 2024a*; *Li et al., 2024b*). Given its recognized biological characteristics, including antioxidant, anti-inflammatory, antibacterial, and antifungal activities, *F. tikoua* has garnered significant attention from pharmacologists. Recent reports suggest that *F. tikoua* may serve as a promising nurse plant for the revegetation and phytostabilization of Pb–Zn tailing wastelands during the initial stages of remediation (*Zhou et al., 2022*). Additionally, its extract shows promise as a treatment for urolithiasis (*Bervinova et al., 2022*), its fruit are a reliable source of functional food with immunoregulatory and antioxidant effects (*Gong et al., 2022*), and it exhibits antidiabetic efficacy both *in vitro* and *in vivo* (*Wang et al., 2022*).

Propagation through stem cuttings is a convenient method that does not require a sterile environment, with low cost and high multiplication coefficient (*Xiao, Niu & Kozai, 2011*). Up to this point, while various approaches have been explored, an efficient method for stem cutting propagation of *F. tikoua* has not yet been conclusively established.

Consequently, *F. tikoua* remains largely in a wild state, lacking proven techniques for variety breeding, artificial cultivation, and seedling propagation, hindering its large-scale development and utilization. Given its remarkable environmental adaptability and vegetative propagation capacity, this study integrated habitat heterogeneity, physiological indicators, and environmental factors to comprehensively analyze the effects of size, age, burial depths, seasonality, substrates, and habitats on the cutting propagation of *F. tikoua*. The goal was to explore the optimal conditions for its commercial-scale reproductive system. This study successfully established an efficient propagation system through cuttings, identifying not only the optimal substrates for cutting propagation from different habitats but also the best stem age, segment size, insertion depth, and month for propagation. These findings provide new references for the establishment of germplasm resource collection, variety breeding, artificial cultivation, and large-scale seedling production of *F. tikoua*.

## MATERIALS AND METHODS

### Test site and environmental conditions

The experiment was conducted annually within the teaching practice facility of Tongren University from 2020 to 2023. Situated in a subtropical monsoon humid climatic zone, the test site experiences variable temperatures and persistent rainfall during spring, intense heat, ample sunlight, and severe drought in the summer, rapid cooling and frequent rainfall in autumn, and low temperatures, scarce sunlight, minimal precipitation, and a brief frost period in winter. The average annual temperature ranges between 15–17 °C, and the average annual precipitation is between 1,100–1,300 mm.

For the propagation *via* cuttings, we selected non-woven fabric pots with dimensions of 18 cm × 18 cm × 20 cm (length × width × height) prior to soil filling. Post soil filling, the dimensions of the pots were 14 cm × 14 cm × 14 cm. Each pot was intended for the insertion of a single stem segment. The cuttings in the pots were placed on seedbeds devoid of arched film or shade nets, thereby allowing the cuttings to be nurtured under natural environmental conditions. Furthermore, each treatment contained 30 stem segments, and each experiment was conducted in triplicate to ensure statistical reliability.

### Methodological assessment of stem segment dimensions

Stem segments of *F. tikoua*, naturally growing in diverse habitats, were sampled from four distinct environmental habitats: grasslands, sandy areas, rocky terrains, and understories. We employed a two-factor factorial design to investigate the influence of stem segment length (L) and diameter (D) on the propagation efficacy of *F. tikoua*. A total of 12 experimental treatments were formulated as follows: H1 ($L = 6$ cm, $D = 0.3 \sim 0.5$ cm), H2 ($L = 8$ cm, $D = 0.3 \sim 0.5$ cm), H3 ($L = 10$ cm, $D = 0.3 \sim 0.5$ cm), H4 ($L = 12$ cm, $D = 0.3 \sim 0.5$ cm), H5 ($L = 6$ cm, $D = 0.5 \sim 0.7$ cm), H6 ($L = 8$ cm, $D = 0.5 \sim 0.7$ cm), H7 ($L = 10$ cm, $D = 0.5 \sim 0.7$ cm), H8 ($L = 12$ cm, $D = 0.5 \sim 0.7$ cm), H9 ($L = 6$ cm, $D = 0.7 \sim 1.0$ cm), H10 ($L = 8$ cm, $D = 0.7 \sim 1.0$ cm), H11 ($L = 10$ cm, $D = 0.7 \sim 1.0$ cm), H12 ($L = 12$ cm, $D = 0.7 \sim 1.0$ cm). The stem segments were pre-soaked in a 0.1% carbendazim solution with a rooting hormone powder at a millesimal concentration for 30 min. Subsequently, they were implanted into nursery bags containing a mixed substrate composition (loam:

light substrate: humus soil in a ratio of 2:1:1). The insertion depth was set to half the length of the stem segment, ensuring that the morphological base of the segment was buried in the soil.

## Influence of insertion depth on stem cutting propagation

To determine the optimal insertion depth for stem cutting propagation, a series of burial depths were established. The basal ends of the stem segments were vertically inserted into seedling bags containing a homogeneous mixture of substrate, with the ratio of loam to light substrate to humus soil being 2:1:1. The insertion depths for the stem segments were categorized as follows: D1 (2~3 cm), D2 (3~5 cm), D3 (5~7 cm), D4 (7~9 cm). Given the prevalence of buds measuring 10 cm in length with a diameter of 0.5~0.7 cm in the stem propagation system of *F. tikoua*, and considering the higher shoot induction rates observed in preliminary experiments, stem segments of 10 cm in length and 0.5~0.7 cm in diameter were selected as the experimental material.

## Influence of stem age on propagation success

The effects of different stem ages on buds, shoots and roots induction rate were studied to identify the best ages of stem segments of *F. tikoua* from different habitats. Stem segments, each measuring 10 cm in length and with a diameter of 0.5~0.7 cm, were harvested from *F. tikoua* plants categorized into four distinct age groups: A1 (1~2 years), A2 (2~3 years), A3 (3~4 years), A4 (4~5 years). For the experimental setup, only stem segments with an insertion depth of 5~7 cm was selected to ensure consistency and comparability across the different age groups.

## Influence of seasonal variations on seedling emergence rates

To assess the impact of distinct seasonal periods on seedling formation rate, a longitudinal study was conducted from 2020 to 2023. For this investigation, stem segments aged of 2~3 years were chosen to ascertain the optimal months for the propagation of *F. tikoua* cuttings sourced from four different habitats. The cuttings were inserted in February, March, April, May, August, September, October, and November, and the resultant plants were cultivated for a continuous period of 8 weeks to evaluate their growth and establishment.

## Influence of substrate composition on bud, shoot, and root induction

To evaluate the influence of different substrate compositions on the induction of buds, shoots, and roots. A series of six substrate blends were prepared based on volumetric ratios of loam, light substrate, and humus soil: T1 (loam: light substrate: humus soil = 1:0:0), T2 (loam: light substrate: humus soil = 1:1:0), T3 (loam: light substrate: humus soil = 1:1:1), T4 (loam: light substrate: humus soil = 2:1:1), T5 (loam: light substrate: humus soil = 2:2:1), and T6 (loam: light substrate: humus soil = 2:3:1). The experimental material consisted of segments with length of 10 cm, diameter of 0.5~0.7 cm, insertion depth of 5~7 cm, and an age of 2~3 years. These experiments were conducted between March and May.

## Parameter determination

For each treatment in these experiments, we quantified the number of buds, shoot length and diameter, and the number and length of roots longer than 0.2 cm on individual cuttings. Buds exceeding a length of 2.0 cm were designated as shoots. The diameter of shoots was determined by measuring the widest part of a single shoot with a precision of 0.01 mm. These measurements were taken at the conclusion of an 8-week period following cutting propagation. The induction rates of buds, shoots, and roots, the root formation rate, and the multiplication coefficients were calculated using the following formulas (*Zou et al., 2022*).

Bud Induction Rate (%) = (Number of budding segments/Total number of segments) × 100.

Shoot Induction Rate (%) = (Number of shooting segments/Total number of segments) × 100.

Root Induction Rate (%) = (Number of rooting segments/Total number of segments) × 100.

Net Multiplication Coefficient = (Total number of buds, shoots or roots)/(Number of sprouting segments).

Total Multiplication Coefficient = (Total number of buds, shoots or roots)/(Total number of segments).

Seedling Formation Rate (%) = (Number of shooting segments with roots/Total number of segments) × 100.

## Statistical analysis

The experimental data were subjected to rigorous statistical analysis using Microsoft Excel 2016 (Microsoft, Bellingham, WA, USA) and SPSS 25.0 software (IBM, Armonk, NY, USA). For comparisons among three or more groups, the Duncan multiple range test was employed to evaluate the significance of differences. When comparing two groups, independent samples t-tests were utilized. GraphPad Prism 5 and Canvas X software were selected for the generation of graphical representations. Furthermore, the influence of habitats on functional traits were examined by employing the Mantel test correlation, as implemented in the "ggcor" package within the R software environment (*Zi et al., 2024*).

# RESULTS

## Effects of stem segment dimensions on shoot induction rates and total multiplication coefficient

Following a continuous 8-week cultivation period of *F. tikoua* cuttings obtained from different habitats, the number of shoots per cutting was enumerated (Table 1). Our statistical analysis revealed that both the stem segment length and diameter exerted significant influences on the rate of shoot induction and the total multiplication coefficient (TMC). The lowest induction rate of shoots was recorded at $24.40 \pm 6.90\%$ in Test H1-understory ($L = 6$ cm, $D = 0.3{\sim}0.5$ cm). Conversely, the highest induction rate, at $92.20 \pm 5.10\%$, was observed in Test H8-sandy ($L = 12$ cm, $D = 0.5{\sim}0.7$ cm). The total

Tan et al. (2024), *PeerJ*, DOI 10.7717/peerj.18768

**Table 1 Influence of segment length and diameter on shoot induction and total multiplication coefficient in *F. tikoua* from diverse habitats.**

| Test | Shoot Induction Rate (%) | | | | Total Multiplication Coefficient | | | |
|------|-----------|-------|-------|------------|-----------|-------|-------|------------|
| | Grassland | Sandy | Rocky | Understory | Grassland | Sandy | Rocky | Understory |
| H1 | 37.80 ± 10.70[C,a] | 43.30 ± 6.70[C,a] | 31.10 ± 5.10[B,a] | 24.40 ± 6.90[C,a] | 0.47 ± 0.12[E,a] | 0.71 ± 0.08[D,a] | 0.46 ± 0.07[C,a] | 0.42 ± 0.08[D,a] |
| H2 | 47.80 ± 8.04[C,ab] | 53.30 ± 10.00[BC,a] | 37.80 ± 5.10[B,ab] | 31.10 ± 5.10[C,b] | 0.69 ± 0.07[DE,ab] | 0.92 ± 0.17[CD,a] | 0.64 ± 0.12[C,ab] | 0.58 ± 0.08[CD,b] |
| H3 | 62.20 ± 5.10[BC,a] | 66.70 ± 3.30[B,a] | 54.40 ± 5.10[AB,ab] | 41.10 ± 5.10[BC,b] | 0.92 ± 0.05[D,a] | 1.14 ± 0.08[C,a] | 0.96 ± 0.08[B,a] | 0.89 ± 0.12[BC,a] |
| H4 | 54.40 ± 5.10[BC,a] | 55.60 ± 5.10[BC,a] | 37.80 ± 6.90[B,ab] | 24.40 ± 8.40[C,b] | 0.71 ± 0.04[DE,b] | 1.01 ± 0.13[C,a] | 0.74 ± 0.10[BC,ab] | 0.64 ± 0.08[CD,b] |
| H5 | 45.60 ± 5.10[C,a] | 61.10 ± 10.20[BC,a] | 44.40 ± 5.10[B,a] | 57.80 ± 5.10[B,a] | 0.94 ± 0.12[D,a] | 1.03 ± 0.15[C,a] | 0.89 ± 0.16[BC,a] | 0.82 ± 0.12[C,a] |
| H6 | 63.30 ± 6.70[BC,a] | 68.90 ± 10.70[B,a] | 62.20 ± 5.10[AB,a] | 64.40 ± 1.90[AB,a] | 1.38 ± 0.20[BC,ab] | 1.51 ± 0.27[B,a] | 1.08 ± 0.17[AB,b] | 1.12 ± 0.11[B,b] |
| H7 | 91.10 ± 6.90[A,a] | 84.40 ± 11.70[AB,ab] | 71.10 ± 5.10[A,b] | 72.20 ± 5.10[AB,ab] | 2.02 ± 0.22[A,a] | 1.91 ± 0.15[A,a] | 1.33 ± 0.12[A,b] | 1.41 ± 0.18[AB,b] |
| H8 | 71.10 ± 6.90[B,b] | 92.20 ± 5.10[A,a] | 65.60 ± 6.90[A,b] | 81.10 ± 5.10[A,ab] | 1.61 ± 0.17[B,b] | 2.15 ± 0.18[A,a] | 1.04 ± 0.10[AB,c] | 1.65 ± 0.11[A,b] |
| H9 | 37.80 ± 8.40[C,a] | 42.20 ± 8.40[C,a] | 26.70 ± 6.70[B,a] | 31.10 ± 8.40[C,a] | 0.71 ± 0.08[DE,ab] | 0.91 ± 0.12[CD,a] | 0.51 ± 0.15[C,b] | 0.44 ± 0.13[D,b] |
| H10 | 48.90 ± 6.90[C,ab] | 57.80 ± 8.40[BC,a] | 34.40 ± 5.10[B,b] | 47.80 ± 3.80[BC,ab] | 1.03 ± 0.07[CD,ab] | 1.14 ± 0.13[C,a] | 0.84 ± 0.22[BC,b] | 0.78 ± 0.19[C,b] |
| H11 | 67.80 ± 5.10[BC,ab] | 75.60 ± 5.10[AB,a] | 52.20 ± 5.10[AB,b] | 57.80 ± 5.10[B,ab] | 1.25 ± 0.10[C,ab] | 1.33 ± 0.19[BC,a] | 0.97 ± 0.07[B,b] | 1.13 ± 0.10[B,ab] |
| H12 | 57.80 ± 5.10[BC,a] | 55.60 ± 5.10[BC,a] | 42.20 ± 6.90[B,a] | 46.70 ± 6.70[BC,a] | 0.82 ± 0.08[D,a] | 0.89 ± 0.08[CD,ab] | 0.76 ± 0.08[BC,a] | 1.02 ± 0.14[BC,a] |

**Notes.**

Data are represented as mean ± standard deviation (SD).

Capital letters denote significant differences among various segment size treatments ($P \leq 0.05$), while lowercase letters signify significant differences across habitats ($P \leq 0.05$), as determined by Duncan's multiple range test.

multiplication coefficient for shoot production was found to be the lowest at $0.42 \pm 0.08$ in Test H1-understory ($L = 6$ cm, $D = 0.3 \sim 0.5$ cm). In contrast, the highest TMC value, $2.15 \pm 0.18$ was noted in Test H8-sandy ($L = 12$ cm, $D = 0.5 \sim 0.7$ cm).

Despite variations in the extreme values, the trends in shoot induction rate and TMC showed parallels across the four different habitats (Table 1). At a given constant diameter or length, both the shoot induction rates and TMC exhibited an initial increase followed by a decrease with an increase in segment size. Although the shoot induction rate and TMC in Test H8 reached the highest values in both sandy and understory habitats, these values were not significantly different compared to those of Test H7. In summary, the segments in Test H7 ($L = 10$ cm, $D = 0.5 \sim 0.7$ cm) consistently demonstrated superior shoot induction rate and TMC across the four distinct habitats. Consequently, the stem segments with a length of 10 cm and a diameter of $0.5 \sim 0.7$ cm from four habitats were chosen for subsequent treatments.

### Effects of segment insertion depth on propagation

Our findings indicate that varying depths of segment insertion significantly influenced the root production of *F. tikoua*. The data conclusively demonstrated that deeper insertion depths were associated with higher root production, as presented in Table 2. The highest rates of root induction and the net multiplication coefficients (NMC) were observed in Test D4 ($7 \sim 9$ cm), whereas the lowest values were found in Test D1 ($2 \sim 3$ cm). With the exception of grassland, the root induction rates in Test D3 ($5 \sim 7$ cm) across the remaining three habitats did not differ significantly from Test D4. However, NMC of root production in Test D4 were substantially higher compared to the other three tests, irrespective of habitat. Conversely, in contrast to the root production data, the shoot induction rate and NMC exhibited a trend of initial increase followed by a decrease with increasing insertion depth, with the lowest values occurring in Test D1 and the highest in Test D3 ($5 \sim 7$ cm). The values in Test D3 were significantly elevated compared to the other three tests across all habitats. Although the four different habitats displayed similar trends in both root and shoot development, significant discrepancies between Test D3 and D4 complicate the determination of an optimal insertion depth based solely on these parameters.

It is noteworthy that each stem segment of *F. tikoua* possesses numerous potential adventitious growth points capable of differentiating into either adventitious roots or shoots. Shallow insertion leads to rapid water loss and segment inactivation, with roots primarily forming at the base of the segment, resulting in limited root and shoot production, and a diminished multiplication coefficient rate, as detailed in Table 2. On the other hand, excessively deep insertion can impair respiration, hindering the emergence of new shoots even when bud primordia have developed. The simultaneous presence of roots and shoots is critical for successful stem cutting propagation. Consistent with the trends in shoot production and root NMC, the highest seedling formation rates were achieved with Test D3 (Table 2). Consequently, an insertion depth of $5 \sim 7$ cm for stem segments from four habitats were deemed optimal for subsequent treatments.

**Table 2** Influence of segment insertion depth on induction, multiplication, and seedling formation in *F. tikoua* across habitats.

| Habitats | Depth Test | Root Production | | Shoot Production | | Seedling Formation Rate (%) |
|---|---|---|---|---|---|---|
| | | Induction Rate (%) | Net Multiplication Coefficient | Induction Rate (%) | Net Multiplication Coefficient | |
| Grassland | D1 (2~3 cm) | 64.44 ± 10.72[c] | 3.60 ± 0.35[b] | 47.78 ± 6.94[d] | 2.44 ± 0.26[c] | 24.44 ± 5.09[d] |
| | D2 (3~5 cm) | 75.56 ± 6.94[b] | 4.23 ± 0.13[b] | 61.11 ± 8.39[c] | 3.56 ± 0.23[b] | 47.78 ± 5.09[c] |
| | D3 (5~7 cm) | 84.44 ± 11.71[b] | 6.37 ± 0.30[a] | 92.22 ± 6.94[a] | 4.61 ± 0.55[a] | 92.22 ± 6.94[a] |
| | D4 (7~9 cm) | 94.44 ± 5.09[a] | 7.19 ± 0.33[a] | 78.89 ± 3.84[b] | 3.20 ± 0.34[bc] | 71.11 ± 5.09[b] |
| Sandy | D1 (2~3 cm) | 77.17 ± 8.39[b] | 3.51 ± 0.14[c] | 51.11 ± 6.94[c] | 2.64 ± 0.39[c] | 26.67 ± 3.33[d] |
| | D2 (3~5 cm) | 78.89 ± 6.94[b] | 4.22 ± 0.09[c] | 74.44 ± 5.09[b] | 3.68 ± 0.24[b] | 52.22 ± 6.94[c] |
| | D3 (5~7 cm) | 87.78 ± 6.94[ab] | 5.35 ± 0.19[b] | 92.22 ± 6.94[a] | 5.75 ± 0.62[a] | 91.11 ± 5.09[a] |
| | D4 (7~9 cm) | 95.56 ± 3.85[a] | 7.49 ± 0.52[a] | 81.11 ± 7.70[b] | 4.60 ± 0.27[b] | 73.33 ± 5.77[b] |
| Rocky | D1 (2~3 cm) | 57.78 ± 5.09[c] | 3.42 ± 0.17[c] | 38.89 ± 5.09[d] | 1.91 ± 0.16[b] | 22.22 ± 5.09[d] |
| | D2 (3~5 cm) | 68.89 ± 5.09[b] | 3.98 ± 0.40[c] | 54.44 ± 8.39[c] | 2.50 ± 0.18[ab] | 33.33 ± 3.33[c] |
| | D3 (5~7 cm) | 77.78 ± 3.85[ab] | 5.40 ± 0.13[b] | 88.89 ± 5.09[a] | 2.96 ± 0.42[a] | 84.44 ± 5.09[a] |
| | D4 (7~9 cm) | 85.56 ± 5.09[a] | 6.53 ± 0.79[a] | 68.89 ± 5.09[b] | 1.92 ± 0.06[b] | 66.67 ± 3.33[b] |
| Understory | D1 (2~3 cm) | 65.56 ± 5.09[c] | 3.13 ± 0.25[d] | 37.78 ± 5.09[d] | 1.78 ± 0.15[b] | 23.33 ± 3.33[d] |
| | D2 (3~5 cm) | 74.44 ± 5.09[bc] | 4.28 ± 0.16[c] | 57.80 ± 8.39[c] | 2.29 ± 0.02[b] | 38.89 ± 5.09[c] |
| | D3 (5~7 cm) | 81.11 ± 5.09[ab] | 5.56 ± 0.35[b] | 91.11 ± 6.94[a] | 3.46 ± 0.27[a] | 82.22 ± 3.85[a] |
| | D4 (7~9 cm) | 88.89 ± 5.09[a] | 6.95 ± 0.52[a] | 72.22 ± 5.09[b] | 2.40 ± 0.46[b] | 65.56 ± 3.85[b] |

**Notes.**

Data are presented as mean ± standard deviation (SD).

Different lowercase letters denote significant differences ($P \leq 0.05$) among depths within the same habitat based on Duncan's multiple range test.

## Effects of segment age on propagation

The age of stem segments significantly influenced root and shoot production, specifically the induction rates and the number of roots and shoots (Table 3). Our findings revealed that the root induction rate and NMC of stem segments from A2 (2~3 years) were significantly higher than those from groups A1 (1~2 years), A3 (3~4 years), and A4 (4~5 years) ($P < 0.05$). Conversely, the shoot induction rate of stem segments from groups A2 and A3 were significantly higher than those from groups A1 and A4 ($P < 0.05$), with no significant differences observed between A2 and A3 ($P > 0.05$). Notably, the shoot NMC for segments from A2 was significantly higher than the other age groups tested.

Considering the substantial decline in NMC for both roots and shoots among segments from groups A1, A3, and A4 (Table 3), segments from A2, which exhibited the highest production of roots and shoots, are likely the most optimal candidates for further study.

In addition to evaluating roots and shoots, we also statistically analyzed the seedling formation rate (Table 3). The data indicated that at the age of 1~2 years, seedling formation was rare, with rates below 42%. In contrast, segments aged 2~3 years consistently produced a significant number of seedlings, with the highest formation rates were observed, ranging from 83.33% in understory habitat to 93.33% in sandy habitat. However, at the age of 3~4 years, seedling formation rates decreased markedly, falling below 80% across all habitats. Most notably, for segments aged 4~5 years, seedlings formation rates dropped below 70% in grassland and sandy habitats, and below 50% in rocky and understory habitats. Given the similar trend among segments from four different habitats in response to age, and

**Table 3** Influence of segment age on induction rate, multiplication coefficient, and seedling formation in *F. tikoua* across diverse habitats.

| Habitats | Ages Test | Root Production | | Shoot Production | | Seedling Formation Rate (%) |
|---|---|---|---|---|---|---|
| | | Induction Rate (%) | Net Multiplication Coefficient | Induction Rate (%) | Net Multiplication Coefficient | |
| Grassland | A1 (1~2 years) | $53.33 \pm 6.67^c$ | $2.13 \pm 0.08^c$ | $45.56 \pm 1.92^c$ | $2.26 \pm 0.22^c$ | $41.11 \pm 6.94^d$ |
| | A2 (2~3 years) | $92.22 \pm 5.09^a$ | $7.44 \pm 0.39^a$ | $93.33 \pm 3.33^a$ | $5.66 \pm 1.24^a$ | $92.22 \pm 5.09^a$ |
| | A3 (3~4 years) | $78.89 \pm 6.94^b$ | $4.21 \pm 0.11^b$ | $87.78 \pm 1.92^{ab}$ | $4.22 \pm 0.48^b$ | $78.89 \pm 8.39^b$ |
| | A4 (4~5 years) | $71.11 \pm 6.94^b$ | $3.11 \pm 0.06^c$ | $78.89 \pm 5.09^b$ | $3.17 \pm 0.80^c$ | $66.67 \pm 6.67^c$ |
| Sandy | A1 (1~2 years) | $41.11 \pm 5.09^d$ | $2.23 \pm 0.60^c$ | $36.67 \pm 3.33^c$ | $1.99 \pm 0.41^c$ | $37.78 \pm 3.85^d$ |
| | A2 (2~3 years) | $93.33 \pm 3.33^a$ | $7.77 \pm 0.36^a$ | $95.56 \pm 1.92^a$ | $6.57 \pm 1.05^a$ | $93.33 \pm 3.33^a$ |
| | A3 (3~4 years) | $82.22 \pm 5.09^b$ | $5.28 \pm 0.24^b$ | $86.67 \pm 3.33^{ab}$ | $5.45 \pm 1.14^b$ | $77.78 \pm 5.09^b$ |
| | A4 (4~5 years) | $58.89 \pm 5.09^c$ | $2.96 \pm 0.18^c$ | $77.78 \pm 3.85^b$ | $2.23 \pm 0.48^c$ | $67.78 \pm 5.09^c$ |
| Rocky | A1 (1~2 years) | $24.44 \pm 6.94^d$ | $1.86 \pm 0.24^c$ | $21.11 \pm 5.09^c$ | $1.47 \pm 0.06^b$ | $21.11 \pm 6.94^d$ |
| | A2 (2~3 years) | $86.67 \pm 6.67^a$ | $5.05 \pm 0.38^a$ | $87.78 \pm 5.09^a$ | $3.36 \pm 0.56^a$ | $86.67 \pm 6.67^a$ |
| | A3 (3~4 years) | $67.78 \pm 6.94^b$ | $3.66 \pm 0.20^b$ | $81.11 \pm 1.92^a$ | $1.92 \pm 0.11^b$ | $74.44 \pm 3.85^b$ |
| | A4 (4~5 years) | $38.89 \pm 5.09^c$ | $2.21 \pm 0.17^c$ | $65.56 \pm 5.09^b$ | $1.16 \pm 0.16^b$ | $43.33 \pm 8.82^c$ |
| Understory | A1 (1~2 years) | $36.67 \pm 6.67^d$ | $1.93 \pm 0.17^d$ | $28.89 \pm 6.94^c$ | $1.72 \pm 0.15^b$ | $31.11 \pm 6.94^d$ |
| | A2 (2~3 years) | $88.89 \pm 5.09^a$ | $6.21 \pm 0.15^a$ | $91.11 \pm 1.92^a$ | $4.24 \pm 0.65^a$ | $83.33 \pm 6.67^a$ |
| | A3 (3~4 years) | $74.44 \pm 5.09^b$ | $4.16 \pm 0.10^b$ | $82.22 \pm 3.85^a$ | $3.22 \pm 0.46^b$ | $61.11 \pm 5.09^b$ |
| | A4 (4~5 years) | $62.22 \pm 8.39^c$ | $3.03 \pm 0.09^c$ | $71.11 \pm 5.09^b$ | $2.20 \pm 0.40^c$ | $48.89 \pm 12.62^c$ |

**Notes.**
Data are presented as mean ± standard deviation (SD).
Different lowercase letters denote significant differences ($P \leq 0.05$) among ages within the same habitat based on Duncan's multiple range test.

**Table 4** Monthly average, maximum, and minimum temperatures in Tongren, Guizhou, China: 2020–2023.

| Temperature (°C) | Month | | | | | | | | | | | |
|---|---|---|---|---|---|---|---|---|---|---|---|---|
| | Jan. | Feb. | Mar. | Apr. | May | June | July | Aug. | Sep. | Oct. | Nov. | Dec. |
| Low | 1.5 | 4.5 | 8.7 | 13.7 | 17.3 | 21.4 | 23.8 | 24.1 | 19.9 | 15.6 | 11.3 | 5.3 |
| High | 9.4 | 12.6 | 20.8 | 26.5 | 32.8 | 35.7 | 38.7 | 33.5 | 31.3 | 25.3 | 19.6 | 12.1 |
| Average | 5.5 | 8.3 | 16.4 | 20.8 | 22.6 | 26.5 | 28.9 | 25.8 | 24.5 | 19.5 | 14.5 | 7.6 |

considering the highest production of roots, shoots and seedlings, segments aged of 2~3 years were selected for subsequent treatments.

## Effects of seasonal variability on experimental outcomes

During the experimental period, the minimum average daily temperature at the test site ranged from 1.5 °C in January to 24.1 °C in August, while the maximum average daily temperature varied from 9.4 °C in January to 38.7 °C in July. The mean temperature recorded was 9.4 °C in January and 28.9 °C in July (Table 4). Notably, February and December exhibited the lowest average temperatures, whereas June and July recorded the highest average temperatures. To mitigate the adverse effects of extreme temperatures on experimental outcomes and ensure at least an 8-week cultivation period, February, March, April, May, Augst, September, October, and November were selected as the initiation months for the monthly test, respectively.
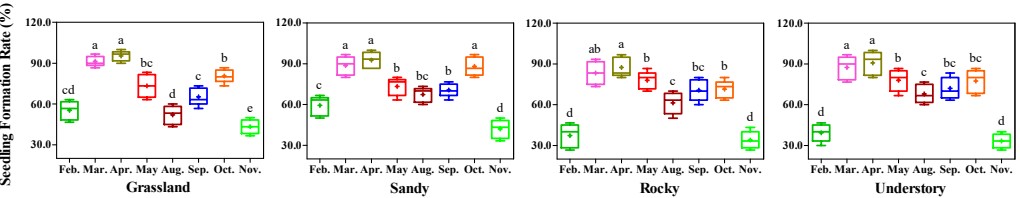

**Figure 1** Effects of different months on seedling formation Rates in *F. tikoua* segments from four distinct habitats. Different letters denote significant differences ($P \leq 0.05$) among month-specific tests within the same habitat, as determined by Duncan's multiple range test.

Significant variation in the seedling production, as determined by seedling formation rate, was observed among the different months tested (Fig. 1). Specifically, cuttings in February and November demonstrated lower seedling formation rates due to the lower average daily temperatures during those months. For May, Augst and September, the seedling formation rates were below 80% across all four habitats, attributed to the elevated average daily high temperatures. Conversely, with favorable daily low temperatures, the highest seedling formation rates were observed in March and April, ranging from 83.34% to 93.33%. Additionally, no significant differences in seedling formation rates were detected between March and April across all tested habitats.

Importantly, the desirable average daily temperatures for cutting propagation led to a noticeable increase in seedling formation rates in October across the four habitats. However, as daily temperatures continue to decline, particularly from November onwards, conducting cutting propagation in October may not be advisable. Consequently, the data from this analysis clearly indicate that March and April are the most suitable months for propagating *F. tikoua* cuttings, and these months were chosen for the subsequent experiments.

## Effects of different substrates on bud productivity

To investigate the influence of various substrates on the bud productivity of *F. tikoua*, cuttings from four different habitats were cultivated continuously for 8 weeks under different soil groups. We analyzed the number of buds germinated per cutting, including both NMC and bud induction rate (Fig. 2). Statistical analysis revealed that among the grassland-derived cuttings, NMC varied in the order of T3 > T4 > T2 > T5 > T1 > T6, with T3 and T4 showing significantly higher values compared to the other four soil groups. This suggests that substrates in T3 and T4 were particularly favorable for bud germination of grassland-derived segments. Notably, T3 and T4 also exhibited significantly higher weekly bud induction rates compared to other soil groups, with no significant differences between each other, further validating the superior performance of these substrates for grassland segments.

In contrast, the substrate in the T5 group was observed to be more advantageous for cuttings from the sandy habitat. However, determining the optimal substrate for bud yield in the rocky habitat was challenging due to the lack of significant differences in NMC
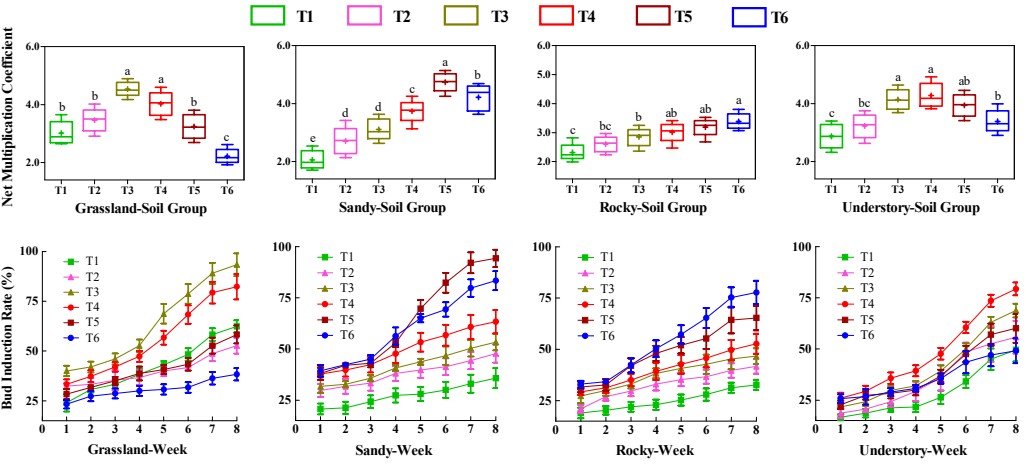

**Figure 2** **Comparative analysis of substrate-induced bud germination in *F. tikoua* cuttings originating from diverse habitats.** Different letters denote statistically significant differences among various soil groups ($P \leq 0.05$) within the same habitat, as determined by Duncan's multiple range test.

among T4, T5, and T6. For the understory habitat, T3, T4, and T5 demonstrated higher NMC without significant differences, while T4 recorded the highest bud induction rate.

Furthermore, the study highlighted the variability in bud productivity and induction across the four habitats within the same soil group (Fig. 2). Cuttings from T1 and T2 yielded fewer buds across four habitats, suggesting that a high loam proportion was detrimental to bud germination, particularly for segments originating from sandy and rocky habitats. Conversely, for grassland and understory habitats, T3 and T4 achieved the highest bud productivity levels. In contrast, T5 and T6 exhibited the highest bud yields and induction rates in sandy and rocky habitats, respectively.

## Effects of different substrates on shoot growth

Following 8 weeks of uninterrupted cultivation, we analyzed the impact of distinct soil types on the shoot growth of *F. tikoua* cuttings sourced from different habitats (Fig. 3). Our findings indicated that the choice of substrate significantly influenced shoot growth. Specifically, the NMC of shoots derived from grassland habitats followed the order T4 > T3 > T5 > T2 > T6 > T1, with T4 exhibiting a statistically significant advantage over the other five groups ($P < 0.05$, Fig. 3A). Furthermore, T4 yielded the longest average shoot length and diameter among grassland-derived cuttings (Figs. 3E and 3I), suggesting that the substrate of T4 was particularly conducive to the growth of shoots from grassland habitats.

In contrast, for cuttings originating from sandy habitats, the highest values of NMC and shoot length were observed in the T6 group, with no significant differences noted when compared to T5 (Figs. 3B and 3F), However, the shoot diameter in T5 was significantly larger than in other soil groups (Fig. 3J). Notably, the T5 group from sandy habitats exhibited the largest shoot diameter across all experimental groups, reaching up to 4.18 mm (Fig. 3J), with no significant differences observed between sandy-T5 and grassland-T4

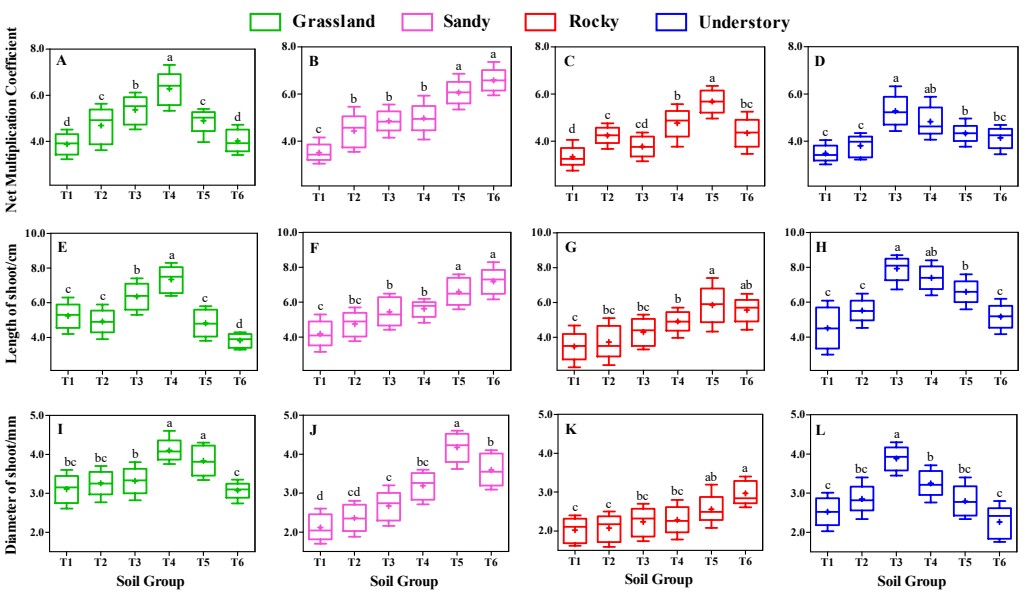

**Figure 3 Comparative analysis of substrate-induced shoot growth in _F. tikoua_ cuttings originating from diverse habitats.** Different lowercase letters indicate significant differences among different soil groups ($P \leq 0.05$) within the same habitat, based on Duncan's multiple range test.

($P > 0.05$, Figs. 3I and 3J). This indicates that cuttings from both sandy and grassland habitats can be readily cultivated to produce thicker shoots, particularly with a slightly higher proportion of light substrates.

For cuttings from rocky habitats, the highest NMC and longest shoots were observed in T5 (Figs. 3C and 3G), while the thickest shoots were found in T6 (Fig. 3K). However, no significant differences in shoot length and diameter were found between T5 and T6 ($P > 0.05$, Figs. 3G and 3K), suggesting that both T5 and T6 were beneficial for shoot growth in rocky habitats. Nevertheless, under the same substrates, the diameter of shoot from rocky habitat was the smallest among the four habitats (Figs. 3I to 3L), suggesting that rocky habitats are less conducive to producing thicker shoots.

Understory habitat-derived cuttings showed peak productivity and shoot size in T3 group (Figs. 3D, 3H and 3L), with T3 and T4 demonstrating comparable levels in terms of NMC and shoot length ($P > 0.05$, Figs. 3D and 3H). Importantly, the longest shoot length in understory-T3 was significantly higher than that in other habitats ($P < 0.05$, Figs. 3E to 3H).

Moreover, different habitats responded variably to shoot growth even under the same substrate conditions. For instance, within the T4 group, the shoot length and diameter followed the sequence grassland > understory > sandy > rocky (Figs. 3E to 3L), indicating that a slightly higher proportion of loam is advantageous for the elongation and thickening of shoots from grassland habitats. In the T3 group, the productivity and size of shoots were ranked as understory > grassland > sandy > rocky (Fig. 3). In the T6 group, the shoot length order was sandy > rocky > understory > grassland (Figs. 3E to 3H), indicating that a higher ratio of light substrate is more favorable for shoot elongation of cuttings from sandy

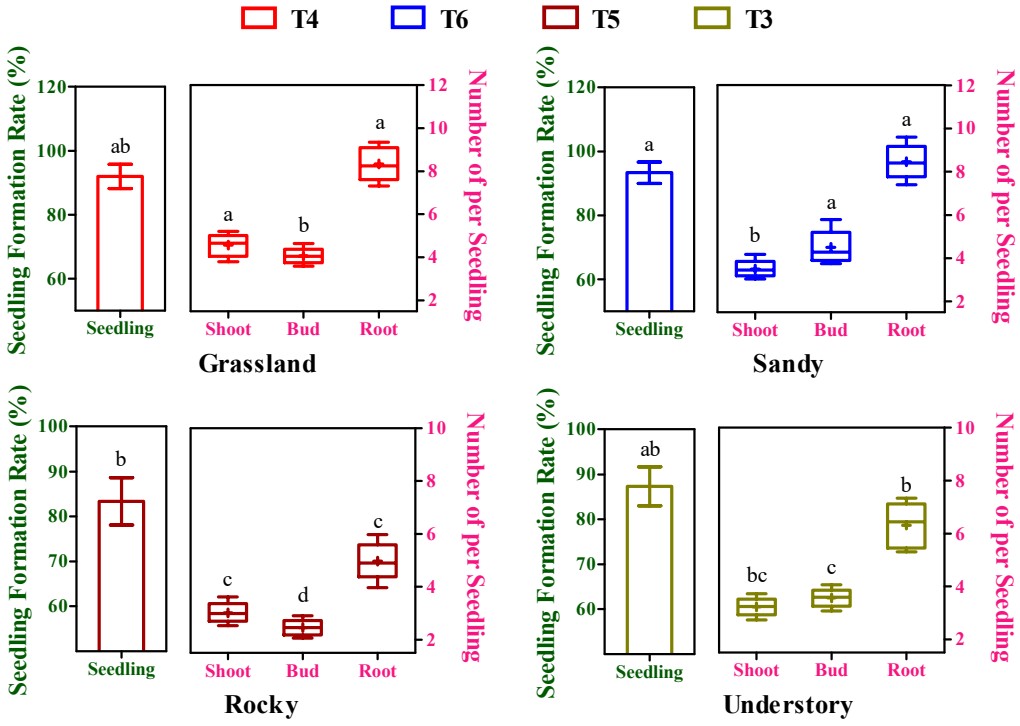

**Figure 4  Effects of different substrates on seedling yield traits of *F. tikoua* cuttings from diverse habitats.** Different lowercase letters indicate significant differences among different habitats ($P \leq 0.05$) for the same target traits (seedling, shoot, bud, and root), as determined by Duncan's multiple range test.

habitats. Additionally, except for T3, T4, T5, and T6, substrates of T1 and T2 resulted in lower levels of productivity and shoot size across the four habitats (Fig. 3), implying that humus soil is essential for the development and growth of *F. tikoua* shoot.

## Effects of different substrates variability on seedling yield

After an 8-week cultivation period under different soil groups, the seedlings, exhibiting simultaneous development of buds, shoots, and roots, were derived from cuttings originating from four distinct habitats and subjected to further statistical analysis. The seedling formation rates exhibited substantial variation among the T3 to T6 soil groups (Fig. 4). Moreover, the highest seedling yield and its corresponding optimal substrate varied across the four habitats (Fig. 4). For instance, the grassland habitat displayed the highest seeding formation rate in the T4 soil group with 92.01%, while the sandy, rocky, and understory habitats peaked in T6 (93.67%), T5 (83.33%), and T3 (87.34%), respectively. Notably, but perhaps the maximum seedling yield in the grassland habitat was significantly higher compared to the rocky habitat ($P < 0.05$, Fig. 4), whereas no significant differences were observed among the grassland, sandy, and understory habitats ($P > 0.05$, Fig. 4).

In a commercial context, the collective influence of budding, shooting, and rooting rates, or overall productivity, is pivotal in determining the propagation viability of a specific clone. To elucidate the impact of variations in budding, shooting, and rooting on overall seedling yield, a composite metric was introduced, representing the mean number

of buds, shoots, and roots generated from seedings derived from the optimal substrate of each habitat (Fig. 4). In terms of shoot productivity, the values across the four habitats ranged from 3.02 to 4.56, with the grassland habitat showing the highest and significantly higher yield compared to other habitats ($P < 0.05$, Fig. 4). Intriguingly, bud productivity exhibited significant variation among the four habitats ($P < 0.05$), peaking in the sandy habitat and being the lowest in the rocky habitat (Fig. 4). Conversely, root productivity varied from 4.97 to 8.46, with grassland and sandy habitats showing comparably higher values, and the rocky habitat having the lowest (Fig. 4). During seedling formation, there appears to be a negative correlation between bud and shoot development, as higher bud yields were accompanied by lower shoot yields, and vice versa (Fig. 4). Furthermore, although there was no significant temporal variation in overall seedling yield among the four habitats, stem segments from the rocky habitat may not be applicable for commercial cutting propagation, given that the seedlings derived from them had the lowest bud, shoot, and root yield.

## Effects of functional traits and environmental factors on cutting propagation

To elucidate the mechanisms underlying the functional characteristics of cutting propagation in *F. tikoua*, we categorized different habitats as distinct matrices to assess their influence on reproductive parameters. By successively conducting Mantel test analyses, we investigated the correlations between functional propagation traits and habitat factors (Fig. 5). Notably, the highest correlation coefficients, approaching 1, were observed between the following pairs: length of root (LR) and bud induction rate (BI), LR and net multiplication coefficient of root (NR), NR and root number per seedling (RN). Except for the net multiplication coefficient of shoots (NS) and length of shoots (LS), the seedling formation rate (SFR) exhibited robust positive correlation with all other ten functional traits. Conversely, shoot number per seedling (SN) demonstrated a significant positive correlation with LR, NR, BI, RN, net multiplication coefficient of bud (NB), and diameter of shoot (DS). In addition to LS, NS and SN, the bud number per seedling (BN) exhibited a robust positive correlation with the remaining nine functional traits. RN, similar to but not identical with BN, showed non-significant positive correlations with NS, NB, LS, and root induction rate (RI). DS only exhibited non-significant positive correlations with NS, and RI. RI, however, exhibited non-significant positive correlations with SN, RN, NS, DS, LS, and NR. Surprisingly, LS only exhibited significant positive correlations with NB and DS. These analyses suggest that overall seedling yield following cutting introduction is primarily influenced by functional traits, including the integration of buds, shoot, and roots. Nevertheless, the combined effects of budding and shooting during the cutting propagation are fundamentally determined by root traits, particularly NR and LR.

Furthermore, we explored the interactions between habitats and functional traits that affect cutting propagation. Interestingly, NR exhibited robust positive correlations with grassland, sandy, and understory habitats, emphasizing the complex interplay between these critical habitat factors (Fig. 5). Importantly, a total of eight traits, including BN, DS, LS, NB, BI, NR, LR, and shoot induction (SI), exhibited significant positive correlations

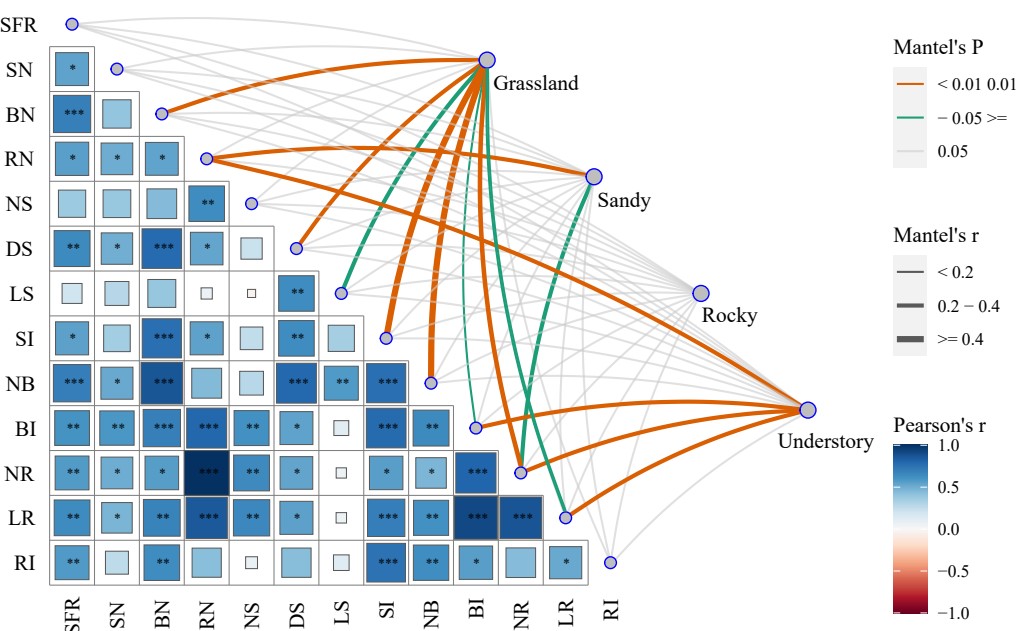

**Figure 5** **Comprehensive analysis of functional traits associated with cutting propagation success of** ***F. tikoua*** **across multiple habitats using Mantel correlation.** The asterisks (*) indicate the level of statistical significance at $P < 0.001$ (three asterisks ***), $P < 0.01$ (two asterisks **), and $P < 0.05$ (single asterisk *), respectively. SFR, Seedling Formation Rate; SN, Shoot Number per Seedling; BN, Bud Number per Seedling; RN, Root Number per Seedling; NS, Net multiplication coefficient of Shoot; DS, Diameter of Shoot; LS, Length of Shoot; SI, Shoot In-duction Rate; NB, Net multiplication coefficient of Bud; BI, Bud Induction Rate; NR, Net multi-plication coefficient of Root; LR, Length of Root in average; RI, Root Induction Rate.

with grassland, reinforcing the optimal stem segment resources for habitat selection in *F. tikoua* cutting propagation. Notably, RN and NR were significantly positively correlated with sandy habitat. Similarly, both RN and NR, along with LR and BI, showed significant positive correlations with understory habitat. These indicate that stem segments from not only grassland but also sandy and understory habitats are suitable for producing seedlings with abundant roots. Finally, the rocky habitat showed non-significant positive correlations with all functional traits, which is consistent with the results of seedling overall yield in rocky habitat (Fig. 5). This suggests that the rocky habitat is an inappropriate source for *F. tikoua* cutting propagation.

## DISCUSSION

*F. tikoua* is a member of the genus *Ficus*, one of the largest genera of angiosperm (*Herre, Jandér & Machado, 2008*). A distinguishing feature of this genus is its possession of a hypanthium, commonly known as the fig, with the bract mouth at the top of the inflorescence serving as the sole conduit for external communication. This unique structure necessitates obligate pollinators for seed production (*Chen et al., 2010*). Consequently, the sexual propagation and dispersal capabilities of *F. tikoua* are significantly constrained by

environmental conditions in its natural habitat. *F. tikoua* is an evergreen, fast-growing, and widely distributed creeping vine, known not only for its landscaping value but also for its effectiveness in slope protection and its promising potential in ecological restoration (*Shi et al., 2012*; *Tian et al., 2020*). Compared to seed-based reproduction, vegetative propagation offers the advantage of shortening the breeding cycle, making it ideal for large-scale propagation while maintaining desirable genetic traits (*Zou et al., 2022*). Thus, a systematic exploration of its cutting propagation technology, based on the principles of asexual reproduction, could provide crucial technical support and theoretical insights for the development and application of *F. tikoua*.

Numerous studies have highlighted the complex temporal and spatial heterogeneity of both essential resources for plant growth and environmental stress factors (*García-Palacios et al., 2012*). Consequently, habitat heterogeneity is a fundamental characteristic of plant natural distribution (*Warren et al., 2014*; *Deák et al., 2021*). *F. tikoua* exhibits significant habitat heterogeneity, adapting to a diverse range of environments including grasslands, sandy soils, rock crevices, riverbanks, and understory conditions (*Deng et al., 2020*). In this study, we observed that the bud, shoot, and root production and growth of *F. tikoua* cuttings from four distinct habitats—grassland, sandy land, rocky land, and understory—demonstrated a degree of habitat-specific variability, even under identical treatment conditions. Additionally, *F. tikoua* is a typical clone plant (*Zhang et al., 2022*), capable of producing new independent individuals through stolons and taproots (*Liu et al., 2011*). Like most other plants, the survival and propagation of *F. tikoua* are influenced by a multitude of factors (*McKey et al., 2010*). Beyond intrinsic functional characteristics, studies have shown that factors such as the size and age of the stem segment, burial method, insertion depth, planting seasonality, and substrates composition can significantly impact the quantity and quality of cutting propagation (*Muñoz Gutiérrez et al., 2009*; *Mabizela, Slabbert & Bester, 2017*). Through two-factor experimental treatments, our results clearly demonstrated the significant effects and optimal selections of size, age, insertion depth, and planting months. Furthermore, we identified irregular but distinguishable effects on the propagation of *F. tikoua* cuttings from the same habitat across different proportions of loam, light substrate, and humus soils, suggesting that both the habitat origin of the stem segments and the soil substrates for cutting propagation should be considered comprehensively.

The hormonal regulation of adventitious root (AR) formation, as discussed in *Lakehal & Bellini (2019)*, is crucial for understanding the propagation success of *F. tikoua* cuttings. As a syconium plant, the propagation of *F. tikoua* primarily relies on stem or stolon propagation to expedite growth and maintain excellent genetic characteristics (*Peng, Compton & Yang, 2010*). The formation of ARs is undeniably critical to the survival of cuttings (*Du et al., 2024*). Enhancing ARs formation ability is essential and can significantly boost the propagation and seedling yield of *F. tikoua*. Our findings revealed that the number of roots in grassland and sandy habitats was notably higher than in other two habitats, suggesting that stem segments from grassland and sandy habitats possess a stronger survival potential due to their higher ARs formation capacity. It has been repeatedly demonstrated that ARs formation in stem cuttings is controlled by various factors, including genetic information

and environmental factors (*Ahkami et al., 2009*; *Steffens & Rasmussen, 2016*), hormonal balance (*Pacurar, Perrone & Bellini, 2014*), and other phytohormones such as abscisic acid (ABA), gibberellins, and indole-3-acetic acid (IAA) (*Huang, Ji & Zhai, 2007*). These hormones, along with redox signaling and intracellular signaling molecules, interact with phytohormone pathways, inducing multiple metabolic alterations during ARs formation (*Druege, Franken & Hajirezaei, 2016*). These metabolic changes, in turn, influence cutting development, such as the successive generation of lateral buds and shoots (*Druege et al., 2019*). Additionally, ARs facilitate gas exchange and enhance water and nutrient uptake during cutting propagation (*Steffens & Rasmussen, 2016*). This dual function supports the development of lateral buds and shoots, which, through photosynthesis, contribute to the autotrophic growth of seedlings.

Root system architecture traits, such as length, diameter, and area, play a pivotal role in determining root performance, enabling plants to acquire water and nutrients, thereby enhancing propagation (*Dodd et al., 2015*). Our Mantel test analyses revealed that NR exhibited a significantly positive correlation with all functional traits except for LS and RI, while LR exhibited a significantly positive correlation with all functional traits except for LS. These findings suggest that LS is not a crucial factor influencing seedling yield in *F. tikoua* cutting propagation. Importantly, NR exhibited a robust positive correlation with RN, LR and BI, and a robust positive correlation was also observed between LR and BI, highlighting consistent relationships within rooting and budding in *F. tikoua* cutting propagation. Additionally, our Mantel test revealed that the functional traits related to cutting propagation parameters—NR, LR, and RN—exhibited significantly positive correlations with grassland, sandy, and understory habitats, respectively, while NR exhibited significantly positive correlations with all three habitats except rocky. This emphasizes the intricate interplay between cutting propagation functional traits and the vital environmental factors of habitats. Optimal root systems support stem growth and enhance seedling resistance, as roots serve as the functional interface between the aboveground part and the soil (*Faget et al., 2013*). Our Mantel test results thus reaffirm the hereditary importance of these relationships between cuttings and habitats. In essence, the functional traits of *F. tikoua* cutting propagation in predominantly depend on the ARs formation ability of stem from different habitats, rather than on the artificial substrates, given its natural habitat openness (*Deng et al., 2020*). Consequently, cuttings from grassland and sandy habitats, which exhibit superior propagation parameters, also demonstrate higher overall seedling yield. From a scientific perspective, soil as an environmental factor plays a critical role in cutting propagation for most plants (*Kontoh, 2016*). Therefore, significant differences in cuttings propagation emerged among different soil groups, despite originating from the same habitats. However, the intrinsic relationship between soil and the functional traits of *F. tikoua* cutting propagation warrants further investigation.

Definitionally, our results suggest that for producing seedlings with a fast germination rate and large number of buds, stem segments from sandy habitat should be selected as cuttings, with a mixed soil of loam, light substrate, and humus soil in a 2:3:1 ratio as the appropriate propagation substrate. For seedlings with longer shoots, stem segments from understory habitat are preferable, with a mixed soil ratio of 1:1:1. Conversely, to obtain

thicker shoots, stem segments from both grassland and sandy habitats are preferred choice, combined with their respective optimal substrates. Additionally, substrate composition significantly impacts root growth, and an appropriate substrate ratio is conducive to root formation, development, and function recovery, thereby promoting shoot growth (*Jaleta & Sulaiman, 2019*). For cuttings with abundant and long roots, stem segments from grassland habitat should be selected, with a mixed soil ratio of 2:1:1 as the propagation substrate.

## CONCLUSION

This study integrated habitat heterogeneity, physiological indicators, and environmental factors to comprehensively evaluate the influence of size, age, burial depths, seasonality, substrates, and habitats on the cutting propagation of *F. tikoua*. Our findings indicate that stem segments of *F. tikoua* with a length of 10 cm and a diameter of 0.5∼0.7 cm exhibit the largest shoot induction rate and total multiplication coefficient. The optimal burial technique involves vertical insertion at a depth of 5∼7 cm. Additionally, stem segments aged 2∼3 years yield the highest seedlings production. The most favorable propagation period for cuttings is between March and April. Furthermore, different substrates demonstrated quantifiable significant effects on the cutting propagation of *F. tikoua* from different habitats. Overall, our results suggest that stem segments sourced from grassland habitat are relatively optimal for *F. tikoua* cutting propagation, with a substrate ratio of loam: light substrate: humus soil = 2:1:1 being particularly advantageous. In contrast, *F. tikoua* from rocky habitat is less suitable for cutting propagation, especially for large-scale seedling production.

### Funding

This work was funded by the National Natural Science Foundation of China (32160086, 32160287), the Projects of Guizhou Provincial Science and Technology (QKHJC-[2019]1455, QKHPTRC-[2020]2003), the Undergraduate Innovation and Entrepreneurship Training Program (grant number S202110665031, S202310665002, S202310665012) and the Central Government Supporting Local Science and Technology Development Fund Project (QKZYD-[2021]4010). The funders had no role in study design, data collection and analysis, decision to publish, or preparation of the manuscript.

### Grant Disclosures

The following grant information was disclosed by the authors:
The National Natural Science Foundation of China: 32160086, 32160287.
Projects of Guizhou Provincial Science and Technology: QKHJC-[2019]1455, QKHPTRC-[2020]2003.
Undergraduate Innovation and Entrepreneurship Training Program: S202110665031, S202310665002, S202310665012.
Central Government Supporting Local Science and Technology Development Fund Project: QKZYD-[2021]4010.

## Competing Interests

The authors declare there are no competing interests.

## Author Contributions

- Tinghong Tan conceived and designed the experiments, analyzed the data, prepared figures and/or tables, and approved the final draft.
- Yu Peng performed the experiments, authored or reviewed drafts of the article, and approved the final draft.
- Biling An performed the experiments, authored or reviewed drafts of the article, and approved the final draft.
- Fan Gao performed the experiments, prepared figures and/or tables, and approved the final draft.
- Yanni Sun performed the experiments, authored or reviewed drafts of the article, and approved the final draft.
- Chuandong Yang conceived and designed the experiments, analyzed the data, authored or reviewed drafts of the article, and approved the final draft.
- Hong Yang conceived and designed the experiments, authored or reviewed drafts of the article, and approved the final draft.
- Zhihong Lu analyzed the data, prepared figures and/or tables, and approved the final draft.

## Data Availability

The raw measurements are available in the Supplemental File.

## Supplemental Information

Supplemental information for this article can be found online at http://dx.doi.org/10.7717/peerj.18768#supplemental-information.

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
