# Peer review of "An efficient propagation system through stem cuttings of a multipurpose plant—Ficus tikoua Bur"

_PeerJ, doi:10.7717/peerj.18768_

## Round 0.1 · original submission · Major Revisions

Dear authors, I ask you to carefully correct all the comments of the reviewers. In Tables 2 and 3, all % (the third, fifth and seventh columns) should be rounded to tenths. The data in Figure 1 should be presented in the form of a box analysis (like Figure 3). I hope for your maximum attention to improving the text, tables and figures.

Reviewer 1 ·

Basic reporting

The manuscript has some issues with grammar, sentence structure, and clarity that need to be addressed for better readability. For instance, I already noticed a sentence that is difficult to interpret in the abstract. "Despite its strong adaptability, rapid growth ability, and widely distribution, however, the current knowledge gap in research on the physiological patterns underlying stem cutting propagation resulting hinders the breakthrough of efficient strategies in commercial-scale reproductive system of F. tikoua., especially its large-scale development is limited."

I recommend that the authors consider revising the manuscript to improve clarity and the overall language quality. Clear and precise language is essential for effectively conveying the research findings.

Experimental design

Please read my comments above.

Validity of the findings

Please read my comments above.

Additional comments

Please read my comments above.

Reviewer 2 ·

Basic reporting

The current paper described propagation system through stem cutting in Ficus tikoua Bur.
Authors performed comprehensively analyze the effects of size, age, developmental stages, substrates and habitats on efficiency of de novo shoot formation.
The text is relatively well organize and data are sound.
English requires minor corrections.

Experimental design

It will be nice to provide some images of de novo shoots for better impression.
Graphs is well-organise. Maybe some font size need to be large (axis name) in some graphs.
Statistical analysis and model is OK.

Validity of the findings

The finding are interesting and can be used in practice.

Additional comments

Some minor points:
line 69: “including India, Vietnam etc” ?? Grammar is not OK these.
There are so much time I swa abbreviations “ F.tikona, in each sentence!
But you work only with one species. So, it can be avoided in majority of the cases.
Line 88: “biological properties” ??
Line 97: process = protocol.
In M&M word “Effects” mean results. It is better to avoid it.
Line 144: “month “ = developmental stage.
Line 180: “statistical” - redundant.
Lines 181- 185: maybe you can organise these numbers as table/graphs?
Line 236: months = developmental stage/status.
Line 308: “seedling productivity” ¿?? Please, use more clear name.
Line 457: seedlings breeding¿¿¿???

Reviewer 3 ·

Basic reporting

Basic report says that there is no knowledge on propagation of Ficus tikoua till data and they are trying establish propagation techniques. But, when literature was searched, there were so many published reports and information on propagation of this species. Lot of tissue culture protocols and stem cutting protocols are available. Also, China as granted patent (No. CN104620827B) on 18th Jan 2018 for the work similar to the present studies which comprises of types of cuttings, number of nodes, time of planting, media standardization etc.
Hence, it may be rewritten about specific condition where this experiment was tried.
English with respect to framing sentences and grammar needs to be rewritten as many of the sentences are not properly written. The manuscript may be referred to referee well versed in English,.

Experimental design

The experiment has designed well with good number of treatments and replications.

Validity of the findings

As lot of reports on similar work are there in public domain, results are not discussed properly with quoting previous works. Also many of the physio biochemical basis of rooting and influence of rooting media needs to be discussed properly.

Additional comments

I suppose, the paper may be rewritten by incorporating above mentioned modification. It may be written as a short communication by clubbing all the findings of each separate experiment. And the final conclusion may be made as the propagation of this crop can be done during this season, using specific size cuttings retaining specific nodes in a specific media. This usual propagation technique (that too many have already reported similar findings) result may not be published as full length paper and it can be a short communication.

---

## Round 0.2 · accepted · Accept

I congratulate you on the acceptance of your article for publication. I hope that your research will help in the widespread introduction of this plant species into various spheres of human activity.

Reviewer 2 ·

Basic reporting

Thanks you for clear corrections. I think that paper can be accepted with minor polishing. My best regards!

Experimental design

All points have been adjusted.

Validity of the findings

Finding were validated.

Additional comments

Papers can be accepted.